# A multicenter analytical performance evaluation of a multiplexed immunoarray for the simultaneous measurement of biomarkers of micronutrient deficiency, inflammation and malarial antigenemia

Eleanor Brindle[1], Lorraine Lillis[2], Rebecca Barney[2], Pooja Bansil[2], Sonja Y. Hess[3], K. Ryan Wessells[3], Césaire T. Ouédraogo[3,4], Francisco Arredondo[5], Mikaela K. Barker[6], Neal E. Craft[7], Christina Fischer[8], James L. Graham[3], Peter J. Havel[3], Crystal D. Karakochuk[6], Mindy Zhang[8], Ei-Xia Mussai[6], Carine Mapango[8], Jody M. Randolph[3], Katherine Wander[9], Christine M. Pfeiffer[8], Eileen Murphy[2], David S. Boyle[2]*

1 Center for Studies in Demography and Ecology, University of Washington, Seattle, Washington, United States of America, 2 PATH, Seattle, Washington, United States of America, 3 Department of Molecular Biosciences, School of Veterinary Medicine and Department of Nutrition, University of California, Davis, California, United States of America, 4 Helen Keller International, Niamey, Niger, 5 Duke University Medical Ctr. Durham, Durham, North Carolina, United States of America, 6 Faculty of Land and Food Systems, University of British Columbia, Vancouver, British Columbia, Canada, 7 Craft Nutrition Consulting, Elm City, North Carolina, United States of America, 8 Centers for Disease Control and Prevention, Atlanta, Georgia, United States of America, 9 Binghamton University (SUNY), Binghamton, New York, United States of America

* dboyle@path.org

**Data Availability Statement:** All relevant data are within the manuscript and its Supporting information files.

## Abstract

A lack of comparative data across laboratories is often a barrier to the uptake and adoption of new technologies. Furthermore, data generated by different immunoassay methods may be incomparable due to a lack of harmonization. In this multicenter study, we describe validation experiments conducted in a single lab and cross-lab comparisons of assay results to assess the performance characteristics of the Q-plex™ 7-plex Human Micronutrient Array (7-plex), an immunoassay that simultaneously quantifies seven biomarkers associated with micronutrient (MN) deficiencies, inflammation and malarial antigenemia using plasma or serum; alpha-1-acid glycoprotein, C-reactive protein, ferritin, histidine-rich protein 2, retinol binding protein 4, soluble transferrin receptor, and thyroglobulin. Validations included repeated testing (n = 20 separately prepared experiments on 10 assay plates) in a single lab to assess precision and linearity. Seven independent laboratories tested 76 identical heparin plasma samples collected from a cohort of pregnant women in Niger using the same 7-plex assay to assess differences in results across laboratories. In the analytical validation experiments, intra- and inter-assay coefficients of variation were acceptable at <6% and <15% respectively and assay linearity was 96% to 99% with the exception of ferritin, which had marginal performance in some tests. Cross-laboratory comparisons showed generally good agreement between laboratories in all analyte results for the panel of 76 plasma specimens, with Lin's concordance correlation coefficient values averaging ≥0.8 for all analytes.

**Funding:** This study received support from the following sources: Author EB received grant # P2C HD042828 from The Eunice Kennedy Shriver National Institute of Child Health and Human Development (https://www.nichd.nih.gov/). This work was supported, in whole or in part, by the Bill & Melinda Gates Foundation [OPP1154343]. DSB received this grant and it also supported LL, RB, PB, and EM in their work. Under the grant conditions of the Foundation, a Creative Commons Attribution 4.0 Generic License has already been assigned to the Author Accepted Manuscript version that might arise from this submission.

**Competing interests:** The authors have read the journal's policy and have the following competing interests to declare: Neal Craft is a paid employee of Craft Nutrition Consulting. This does not alter our adherence to PLOS ONE policies on sharing data and materials. There are no patents, products in development or marketed products associated with this research to declare.

Excluding plates that would fail routine quality control (QC) standards, the inter-assay variation was acceptable for all analytes except sTfR, which had an average inter-assay coefficient of variation of $\geq 20\%$. This initial cross-laboratory study demonstrates that the 7-plex test protocol can be implemented by users with some experience in immunoassay methods, but familiarity with the multiplexed protocol was not essential.

## Introduction

Micronutrient (MN) deficiencies include iron, vitamin A, and iodine amid other essential elements and vitamins [1, 2]. It is estimated that over 2 billion people worldwide are directly affected by a MN deficiency [3]. Children and pregnant women are particularly at risk due to an inadequate diet that fails to meet the greater micronutrient requirements necessary for fetal growth or childhood development [4]. Iron, iodine and vitamin A are three of the micronutrients of greatest public concern [1]. MN deficiency can adversely affect the physiology of diverse organ systems, impairing, for example, ocular, immunologic, and neurological function, often causing irreversible damage [4]. As such the quality of life of those affected by MN deficiency is significantly reduced, making it critical to accurately assess the prevalence of micronutrient deficiency to allow the targeted implementation of micronutrient intervention programs among high risk populations and assess intervention outcomes [5, 6].

Data harmonization for MN deficiency surveillance is challenged by the use of different survey biomarkers and methods by different labs. For example, vitamin A is determined via serum retinol or retinol binding protein 4 (RBP4) which do not always correlate well with each other [7–9], while iodine is measured using urinary iodine, thyroglobulin (Tg) or thyroid hormones [10–12]. Furthermore, the quantitative data generated by enzyme linked immunosorbent assays (ELISAs) is impacted by a variety of factors including the sample type (e.g. dried blood spot [DBS], serum or plasma from venous or capillary blood) [13–15], variations in the antibodies, buffers and protocols used by commercially available ELISA kits [16, 17], a lack of international reference materials for some key biomarkers (e.g. RBP4), and a lack of external quality assessment (EQA) materials to consistently qualify tests and user performance resulting in variation in measurements across laboratories [18]. Finally, in some cases, routinely used immunoassays have not been fully validated by the manufacturer or by users to confirm acceptable assay performance for their intended use [19].

These challenges, either individually or in combination, result in poorer quality datasets that make it difficult to accurately and consistently monitor MN deficiency distribution, prevalence, and severity, in particular, across different surveys using similar but not identical analytical methods. Ideally, MN deficiency data collected in large surveys such as the Demographic and Health Surveys (DHS), which conduct surveillance in different populations globally and at multiple time points, should be uniform to allow constructive comparisons across countries and/or survey waves. A primary purpose of micronutrient status assessments is to understand what populations are most vulnerable and to assess the impact of interventions.

For the accurate inter-region comparison of MN deficiency surveys or along a series of time points, the harmonization of absolute measurements generated via all the analytical methods used is essential. This can be realized, in part, by using inter-laboratory performance studies and evaluations of these methods to identify technologies that are relatively easy to perform and have sufficient accuracy and reproducibility to generate comparable datasets irrespective of where the testing is carried out. We have reported previously on a multiplex assay

method developed to simplify population surveillance of micronutrient status by combining relevant biomarkers into a single test. In this study we report results of a full formal validation of the Q-plex 7-plex Human Micronutrient Array (hereafter the 7-plex), examining the reproducibility observed with multiple users across seven different laboratories in order to characterize measurement variability for biomarkers pertinent to MN deficiency surveillance, namely inflammatory biomarkers alpha-1-acid glycoprotein (AGP) and C-reactive protein (CRP); thyroglobulin (Tg, iodine); serum ferritin and soluble transferrin receptor (sTfR, both iron); retinol binding protein 4 (RBP4, vitamin A); and histidine-rich protein 2 (HRP2, *Plasmodium falciparum* malaria) [9, 11, 20, 21]. We assessed the precision and performance of the 7-plex for use in population surveillance of MN deficiency [22–24].

## Materials and methods

### 7-plex array procedure

The panel samples and controls were thawed on the bench top at room temperature on the day of the assay. Samples were processed following the assay protocol. First, the lyophilized competitor mix provided with the kit was reconstituted in the sample diluent volume recommended in the product insert to produce a 1X strength competitor mix. Next, the lyophilized calibrator was reconstituted with the competitor mix volume indicated in the kit insert, then a series of 7 threefold dilutions was prepared to create an eight-point standard curve. A 15 µL volume of each sample or quality control (QC) was combined with 135 µL competitor mix to produce final dilutions of 1:10. A volume of 50 µL per well of prepared standards, controls and samples were added to the plates in duplicate wells and each plate was incubated at room temperature for 2 hours with shaking on a flatbed shaker at 500 revolutions per minute. All reactions were aspirated, and the wells washed 3 times with the wash buffer provided with the kit. Next, 50 µL of detection mix was added to each well and the plate was then incubated with shaking for 1 hour and then washed one more time as described above. Labeling was performed by adding 50 µL streptavidin horseradish peroxidase solution to each well and shaking for 20 minutes. After washing 6 times, the chemiluminescent substrate mixture of equal volumes of parts A and B were added at 50 µL per well. Each plate was then immediately imaged at 270 seconds of exposure time using a Quansys Q-View™ Imager LS (Quansys Biosciences).

Q-View Software (Quansys Biosciences) was used to overlay a plate map onto the locations of analyte spots in each well to quantify the chemiluminescent signal from each spot in units of pixel intensity. The software applies the calibrator concentration values to the pixel intensities for each spot in the standard curve wells and was set to automatically fit optimal 5 parameter logistic calibration curves for each analyte. The pixel intensities of the spots in each test well were then used to interpolate the concentration of each analyte relative to its calibrator curve. Once the plate image is overlaid with the analysis grid, all of the curve fitting and data reduction steps are automatically applied via the software. The upper and lower limits of quantification determined by Quansys for each kit lot were applied to exclude values beyond the concentration ranges that yield precise concentration estimates.

### Validation of 7-plex performance in a single lab

The intra- and inter-assay performance ranges of the 7-plex reported in earlier publications, along with other components of assay validation, were originally generated by the manufacturer of the assay, Quansys Biosciences [23]. As a follow up to this, before the inter-laboratory evaluation was performed, a second validation of 7-plex performance was conducted independently in the PATH laboratory to confirm the original findings (see Fig 1).

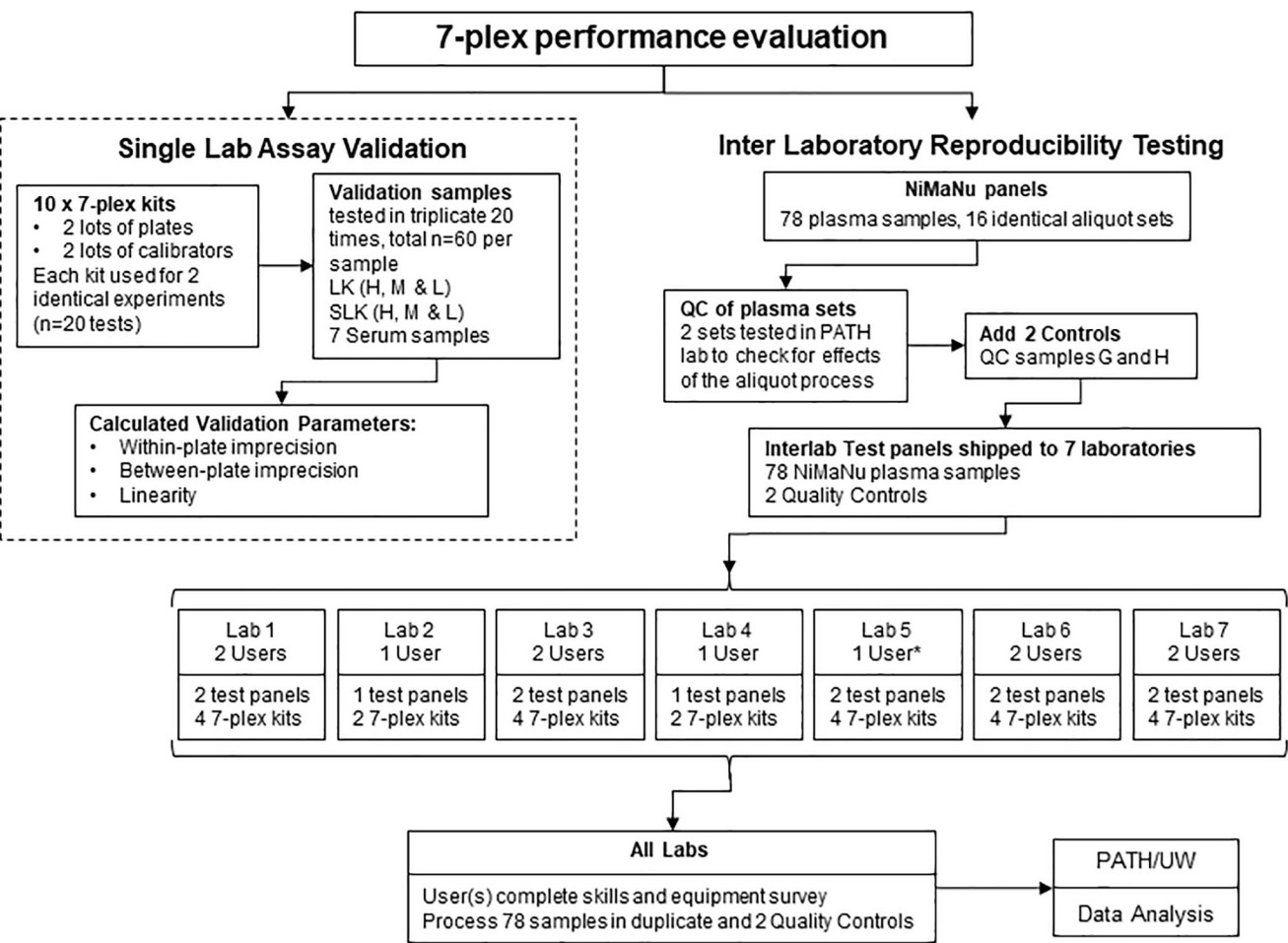

**Fig 1. A flow chart depicting the three primary workstreams followed to prepare and complete the interlaboratory assessment of the 7-plex assay.** This included the validation of two separate lots of both plate and calibrator reagents prior to sharing identical test kits with all partner laboratories; the construction and qualification of the blinded test panels; and finally the inter laboratory assessment of the blinded test panels and analysis of data. * This user ran 4 plates instead of just 2. LK, Liquichek Standard; H, High; M, medium; L, Low; SLK, spiked Liquichek Standard; G and L, Quansys quality control standards; QC, quality control; UW, University of Washington.

**Validation materials.** The test panel used to qualify the 7-plex performance consisted of Liquichek (LK) Immunology Control Level 3 (Lot # 66363, Bio-Rad Laboratories Inc., Hercules, CA, USA), a pooled human serum-based matrix containing most of the analytes of interest. As the LK control has low concentrations of both sTfR and Tg and is negative for HRP2, a spiked version was also prepared by adding concentrated sTfR antigen (Fitzgerald, MA, USA), Human Tg (BiosPacific Inc., Emeryville, CA, USA) and HRP 2 (CTK Biotech, San Diego, CA, USA) to better reflect the quantitative range of the array. Additionally, seven human plasma samples previously determined to have the highest sTfR measurements (all HRP2 negative) were selected from our US donor panel for use in the validation experiments [22, 23].

**Validation experiments.** A flow chart in Fig 1 highlights the sequential processes used to validate the 7-plex assays, construct blinded test panels and finally carry out the inter laboratory assessment. Ten 7-plex plates in total, were employed to evaluate assay precision and linearity. Each plate performed two identical experiments, with all controls and test sample dilutions prepared independently for each experiment, with results derived from an

independent standard curve for each half of a plate. Using a total of 20 replicate experiments, the LK and spiked Liquichek (SLK) were screened in triplicate as high (undiluted), medium (1:4 dilution) and low concentrations (1:10 dilution). The absolute values derived from these samples were used to calculate the independence of volume (linearity) and precision of each assay across its linear dilution range. The seven human plasma specimens were run in duplicate at a 1:10 dilution. To assess real-use imprecision, including common potential sources of variability, the 20 experiments used two different 7-plex plate lots and two different lots of calibrator, with experiments performed by two users. All testing was performed at ambient temperature (approximately 22˚C) following the 7-plex protocol as described in detail below. The ten plates generated 60 unique data points for each biomarker in the LK and SLK dilutions (run in triplicate in 20 experiments) and 40 data points per biomarker in the 7-member plasma panel (run in duplicate in 20 experiments).

**Validation statistical methods.** Validation experiment results were analyzed to estimate intra- and inter-assay imprecision and linearity. A variance components model was used to calculate intra- and inter-assay coefficients of variation (CV) for the LK, SLK and plasma sample results from the 20 independent experiment batches run on ten assay plates [25]. This method parses within-plate variance and between-plate variance to more accurately reflect the sources of variability in imprecision estimates. CVs were calculated with results grouped by plate lot, by calibrator lot, and by user, as well as in aggregate. CV is used to simplify interpretation of estimates of assay variability, but because it is a ratio of standard deviation to mean concentration, it tends to overstate variation at lower concentrations. Linearity was estimated using three dilutions of both the LK and SLK and was calculated by dividing the concentration of a diluted sample by the concentration of the next higher dilution, multiplying by the dilution factor and expressed as a percentage.

## Inter-laboratory assessment

**Donor panel.** While the validation work was carried out with a mixture of serum and plasma samples we have previously demonstrated comparable results between paired serum and heparinized plasma samples [22], thus we were confident plasma samples would be appropriate for interlaboratory assessment. Plasma samples from a study of micronutrient status among pregnant women in Niger were used to generate test panels for the inter-laboratory assessment [23]. Samples were collected as part of a cross-sectional study embedded into the Niger Maternal Nutrition (NiMaNu) Project, which was registered with the U.S. National Institutes of Health (www.ClinicalTrials.gov; NCT01832688) [26]. The National Ethical Committee (Niger) and the Institutional Review Board of the University of California Davis (UC Davis; USA) provided ethical approval for the study protocol and the consent procedure. The local implementation was under the responsibility of Helen Keller International (Niamey, Niger), who followed relevant national regulations and laws applying to project implementation and foreign researchers. Written informed consent was obtained from all study participants.

A total of 18 rural health centers from 2 health districts in the Zinder Region were selected to participate in the NiMaNu project. In each community, pregnant women were randomly selected and invited to participate in the survey. They were eligible if they provided written informed consent, had resided in the village for at least six months, and had no plans to move within the coming two months. As part of the NiMaNu study, venous blood samples were collected and used to prepare heparinized plasma and DBS cards. PATH signed a material transfer agreement (MTA) with UC Davis and then each of the participating laboratories in this study signed an MTA with PATH prior to receipt of test materials.

**Construction of blinded test panels.** The NiMaNu panel of 208 plasma samples were previously analyzed using the 7-plex [23]. As the 7-plex requires only 13.5 μL of plasma per test, multiple samples within this panel had a significant residual volume (>300 μL) of plasma. Using the original 7-plex data, seventy-eight samples with concentrations representing the full range for each analyte were chosen for sub aliquoting to create 16+ identical panels consisting of 78 separate 20 μL plasma samples as follows: The frozen plasma was thawed on ice, spun briefly in a microfuge and pipetted into sterile screw cap tubes, which were then stored -80 ˚C (Fig 1). Two of these samples were randomly chosen from the panel and all 19 of the aliquots prepared from these samples were assessed by the 7-plex to test for tube-to-tube variability that might have been introduced during sub-aliquoting. Both samples had an intra-assay CV < 10% for each analyte (S1 Table), confirming analyte uniformity across sample tubes. The original specimen identifiers of the remaining samples were replaced with sequential numbering from 1–76, effectively blinding the labs previously involved in studies that used specimens from this panel. The samples were stored at -80 ˚C until shipment to the partner laboratories.

In addition to the 76 member Niger heparin plasma panel samples, Quansys Biosciences prepared QC samples, named G and H, representing both high and low analyte values to be run on each plate (Fig 1). These quality controls were used to evaluate whether each plate used during this study would meet acceptance criteria ideally applied in the routine use of the kit. The controls were prepared by spiking serum with purified biomarkers as needed to reach the desired concentration of each biomarker [23]. Prior to distribution, the G and H controls were quantified by Quansys via a series of twenty independent test runs using the 7-plex to determine the expected values of all 7 biomarkers (S2 Table).

**Laboratories.** Seven distinct laboratories offered to be part of the inter-laboratory performance study, each providing data from at least one, and ideally two, laboratorians per facility. Laboratories at PATH, the University of Washington, Quansys, and UC Davis had previously collaborated to develop and verify the performance of the Human Micronutrient assay [22–24] (Fig 1). Other laboratories, including ones from the US Centers for Disease Control and Prevention (CDC, GA), Eurofins Craft Technologies, Inc. (NC), Binghamton University (SUNY), and the University of British Columbia have also been independently evaluating the performance of the Human Micronutrient assay [27–30]. Once each laboratory had signed the MTA to access the samples, two complete sets of 76 heparin plasma samples and two of the G and H quality control sets, were shipped on dry ice via overnight courier. Recipients acknowledged the panels' integrity (frozen with dry ice still in packaging) upon arrival and stored them at -80 ˚C until assay. The manufacturer of the assay, Quansys, was excluded from the study in order to limit bias, as their technical staff are most familiar with the platform and they manufacture and market the Human Micronutrient assay kit. Prior to performing testing, all laboratories were offered a training webinar hosted by an experienced Q-plex user (E. Brindle), to ensure that each study laboratorian was familiar with the test protocol and data analysis methods. All of the array kits used in the inter-laboratory assessment exercise were from the same manufacturing lot. Each plate image was saved and reviewed by an expert user (E. Brindle) to confirm consistency in software settings used to fit calibration curves and report results (Fig 1).

**Assessment of laboratory equipment and user capability to operate the Q-plex assay.** To understand effects of user skills and experience and status of laboratory equipment on results, a questionnaire was distributed prior to testing to collect details from each laboratory. Each operator completed a questionnaire to determine their level of previous experience with the 7-plex, and experience with quantitative immunoassays (Fig 1). An inventory of equipment summarized maintenance histories for items necessary for use with the 7-plex, and specified

the plate washing method. Experience and equipment status questionnaire results were summarized by assigning a scale value to each element, scoring each factor as follows: Lab operator experience (2 elements, 1 to 3 scale, with 3 as most experience), Quansys software experience (0 to 1 scale, 1 is experienced), automated plate washer availability (0 to 1 scale, 1 is available), and recency of calibration (2 elements, 1 to 3 scale with 3 as most recent). Scores were totaled to derive a summary score ranging from 0 (no experience, poor equipment status indicators) to 14 (extensive user experience, all equipment present and recently calibrated).

**Inter-laboratory statistical methods.** Values below the lower limit of quantification (LLOQ) for each analyte were excluded from analyses. Results of the quality control samples run on every plate were evaluated to determine whether the plates would meet acceptance criteria that, for the purposes of this study, were intentionally less stringent than would generally be permitted, whereby at least one control result should have any 6 of the 7 analyte results falling within a 95% confidence interval calculated from all plates in the study. Because the intent of this exercise was to evaluate reproducibility, all plates were included nearly all subsequent analyses. The effect of excluding data from any plates meeting this rejection criteria was considered separately. Inter-assay CV's were calculated to evaluate the performance between the 7 labs and intra-assay CV's were calculated to evaluate the performance within each of the 7 labs. Intra-assay CVs for duplicate wells of the test samples were averaged for each analyte on each plate, and then plate averages were aggregated across analytes to summarize intra-assay CV averages by lab and by operator. Inter-assay CVs were calculated across all plates (n = 12) for each sample (n = 76); inter-assay CVs were then averaged to summarize inter-assay CV for each analyte. Agreement between results across laboratories was assessed using Lin's concordance correlation coefficient (CCC) [31]. Results from assays conducted in the PATH and UW labs by the three operators with the most experience using the 7-Plex were averaged to create a comparison set that was compared to each of the nine remaining assay batches from five labs. Lin's CCC was calculated using STATA version 15.1 (StataCorp, College Station, TX USA).

## Results

### Validation of 7-plex performance in a single lab

All test data can be publicly accessed at Dataverse (https://dataverse.harvard.edu/dataverse/micronutrient_immunoarray). The data derived from 20 independent replicate experiments run on ten 7-Plex assay plates in the PATH lab were used to evaluate the precision (intra- and inter-assay, n = 13 samples) for each assay (see Fig 1). Table 1 provides a summary of results from the validation sample with a value closest to the relevant cutoff concentration for each analyte. The intra-assay CV for each analyte was less than 5%, with the exception of ferritin (5.8%), and all inter-assay CVs were less than 15% (Table 1). These are the accepted maximum CVs for ELISAs and comparable to CVs observed previously in the manufacturer's evaluation of the 7-plex [23, 32]. S3 Table includes all results, including those outside assay limits of quantification; average intra-assay CV was below 5% for all analytes. There was one plasma sample that gave an intra-assay CV of 15.7% with the ferritin assay. However, the concentration in this particular sample was around the LLOQ, and as the generally acceptable threshold at this concentration is 20%, this was still considered acceptable [32]. Average inter-assay CV was ≤15%, with the CV for most samples below 10% for each analyte.

Tests of assay linearity showed no evidence of systemic non-parallelism across dilutions for any analyte. All biomarkers, apart from ferritin, had a linearity of 96% to 99%. The ferritin gave poorer linearity of 57% with SLK samples; however, it was noted that the Tg added to the

**Table 1.  Analytical validation of 7-plex performance.**

|  | AGP (g/L) | CRP (mg/L) | Ferritin (μg/L) | HRP2 (μg/L) | RBP4 (μmol/L) | sTfR (mg/L) | Tg (μg/L)** |
|---|---|---|---|---|---|---|---|
| Calibration range | 0.001–0.37 | 0.028–20.5 | 0.156–114 | 0.001–1.04 | 0.001–1.04 | 0.163–119 | 0.019–13.7 |
| Limits of quantification (mean, n = 20 experiments from 10 plates) | 0.0016–0.354 | 0.0648–20.5 | 0.451–108.3 | 0.0016–0.8865 | 0.005–0.929 | 0.241–118.1 | 0.61–13.7 |
| Optimal cutoff value (1:10 dilution)* | 0.067 | 0.33 | 1.68 | 0.092 | 0.12 | 1.17 | 0.72 |
| Mean QC sample concentration | 0.073 | 0.28 | 0.88 | 0.13 | 0.077 | 1.3 | 0.64 |
| intra-assay %CV | 2.5 | 2.1 | 5.8 | 1.9 | 1.5 | 3.1 | 2.1 |
| inter-assay %CV | 6.5 | 9.1 | 14.3 | 8.8 | 13.6 | 10.0 | 10.8 |
| Linearity (%, LK) | 98 | 98 | 83 | N/A | 99 | 96 | 98 |
| Linearity (%, SLK) | 99 | 99 | 57*** | 99 | 99 | 98 | 98 |

Summary coefficients of variation (control result with mean concentration closest to the relevant cutoff value) and linearity for each analyte. See S3 Table for all results, including those outside the assay limits of quantification.

*Cutoff values estimated by ROC analysis using NiMaNu study classification as a gold-standard; values are given as 1/10 dilution adjusted values to show their relationship to the assay calibration and quantification ranges. CV's calculated using a variance components model to separate within-plate and between-plate contributions to variation.

**Cutoff value estimation confounded by measurement of Tg in DBS in the NiMaNu study which served as the gold-standard for ROC analysis.

***SLK included concentrated Tg from whole blood, which interfered with the ferritin assay.

AGP, α-1-acid glycoprotein; CRP, C-reactive protein; HRP2, histidine rich protein 2; LK, Liquichek; N/A, not available; QC, Quality control; RBP4, retinol binding protein 4; SLK, spiked Liquichek; sTfR, soluble transferrin receptor; Tg, thyroglobulin.

spiked Liquichek was derived and concentrated from whole blood, thus adding this also increased the concentration of ferritin to above the limit of quantification in the high dilution samples. In the normal Liquichek the linearity improved to 83%, though this was still substantially lower than the other linearity values observed. The pooled data presented in Table 1 and S3 Table demonstrates that different operators and/or plate lots did not impact performance. Overall the results confirmed the previously reported findings and the assay was considered suitable for the subsequent inter-lab study [23].

## Inter-laboratory assessment

Eleven operators in seven labs tested the full set of 76 plasma samples with the 7-plex (Table 2). In most cases, each laboratorian tested the entire panel of 76 plasma samples only once (requiring two assay plates per operator). In one laboratory, a single operator assayed the entire panel of 76 plasma samples twice (for a total of four assay plates). In four laboratories, two different users tested the complete panel. Overall, each specimen was tested in duplicate wells 12 times (i.e. 24 data points). All laboratories completed testing within 3 months of each other, and samples were kept frozen until the day of assay. While many of the partners had very limited experience with the Q-plex platform, their laboratorians did have variable levels of experience in performing other immunoassays. All laboratories had the required equipment, including multichannel pipettors and rotating plate shakers. All labs but two had an automated plate washer; the remaining labs used the manual plate washing protocol described by Quansys in the kit instructions for use. Each laboratory had a Q-View imager and analysis software necessary for reading the 7-plex plates and for processing the raw data into concentration values for each analyte. Scores were tallied with overall scores for each lab/laboratorian shown in Table 2. One lab had the maximum possible score of 14 indicating a highly

**Table 2. Average intra-assay (within-plate) and inter-assay (between plate) %CVs by laboratory and operator.**

**Average intra-assay (within plate) %CVs, all samples**

| Laboratory Identifier | 1 | | 2 | 3 | | 4 | 5 | 6 | | 7 | |
|---|---|---|---|---|---|---|---|---|---|---|---|
| n valid results, all samples, all plates, all assays, all operators | 892 | | 431 | 932 | | 454 | 889 | 982 | | 859 | |
| Average intra-assay (within plate) %CV | 3.3 | | 4.7 | 4.4 | | 3.1 | 4.4 | 4.3 | | 7.3 | |
| Operator ID (up to 2 per lab) | 1 | 2 | 3 | 4 | 5 | 6 | 7 | 8 | 9 | 10 | 11 |
| 7-plex assay plates used (n) | 2 | 2 | 2 | 2 | 2 | 2 | 4 | 2 | 2 | 2 | 2 |
| Experience and equipment score (0 to 14 scale, ideal score = 14) | 13 | 12 | 13 | 11 | 13 | 14 | 13 | 7 | 6 | 12 | 8 |
| n valid results, all assays, all plates, all samples run by all operators | 455 | 437 | 431 | 473 | 459 | 454 | 889 | 503 | 479 | 467 | 392 |
| Average intra- assay (within plate) %CV for each operator | 3.2 | 3.3 | 4.7 | 4.5 | 4.4 | 3.1 | 4.4 | 4.3 | 4.3 | 5.7 | 8.8 |

**Average inter-assay (between plate) %CVs, quality control samples**

QC sample G

| Laboratory Identifier | | | 1 | 2 | 3 | 4 | 5 | 6 | 7 |
|---|---|---|---|---|---|---|---|---|---|
| Analyte | Mean conc. | Valid results | | | | | | | |
| AGP | 0.876 g/L | 24 | 6.7 | 2.7 | 9.1 | 1.7 | 10.6 | 27.3 | 38.0 |
| CRP | 10.94 mg/L | 24 | 15.0 | 1.1 | 18.8 | 0.7 | 8.4 | 14.7 | 37.8 |
| Ferritin | 286.62 µg/L | 24 | 8.9 | 4.2 | 5.5 | 13.9 | 5.9 | 6.8 | 8.1 |
| HRP2 | negative | - | - | - | - | - | - | - | - |
| RBP4 | 1.875 µmol/L | 21 | 2.9 | 14.1 | 40.3 | 5.9 | 13.4 | 13.2 | 8.4 |
| sTfR | 11.215 mg/L | 22 | 11.8 | no data | 8.1 | 5.0 | 16.4 | 16.9 | 45.4 |
| Tg | 6.10 µg/L | 24 | 5.3 | 5.5 | 11.1 | 3.7 | 12.0 | 14.8 | 26.2 |
| Inter-assay %CV average | | | 8.4 | 5.5 | 18.1 | 5.1 | 11.1 | 16.6 | 27.3 |
| **QC sample H** | | | | | | | | | |
| AGP | 0.325 g/L | 24 | 8.5 | 1.3 | 7.2 | 0.0 | 11.7 | 6.8 | 24.5 |
| CRP | 1.105 mg/L | 24 | 11.7 | 8.2 | 15.1 | 1.9 | 15.2 | 20.7 | 6.8 |
| Ferritin | 17.767 µg/L | 23 | 2.5 | 0.3 | 6.9 | no data | 10.5 | 12.9 | 47.6 |
| HRP2 | 0.216 µg/L | 22 | 1.3 | 0.0 | 10.9 | no data | 11.2 | 2.0 | 17.0 |
| RBP4 | 0.435 µmol/L | 24 | 18.1 | 17.0 | 4.0 | 2.1 | 16.4 | 4.0 | 19.3 |
| sTfR | 4.327 mg/L | 17 | 2.4 | no data | 15.0 | 21.9 | 14.7 | 12.0 | 60.2 |
| Tg | 1.00.96 µg/L | 21 | 11.2 | 7.6 | 11.7 | no data | 10.0 | 19.4 | 26.8 |
| Inter-assay %CV average | | | 8.0 | 5.7 | 10.1 | 6.5 | 12.8 | 11.1 | 28.9 |
| Mean inter-assay %CV (QC samples) | | | 8.2 | 5.6 | 14.1 | 5.8 | 12.0 | 13.9 | 28.1 |

Intra-assay %CVs are calculated by averaging the well-to-well %CVs from all valid results, all plates run by each laboratory/operator. Inter-assay %CVs are mean values derived from pooling quality control outputs from samples G and H run in duplicate on every plate. The %CVs were calculated using all results that were within the assay limits of quantification. Numbers of possible valid results vary because laboratories ran different total numbers of plates. N is the number of valid results.

experienced laboratorian with access to recently calibrated equipment, while the minimum observed score was 6, indicating a laboratorian that had limited experience with ELISAs, who was not familiar with the Q-view software, and did not have access to a plate washer.

A G- and H-quality control sample were included in duplicate on every plate (Fig 1), with results summarized in Table 2. Multiple analyte results for both controls on one plate were outside the 95% confidence interval derived from all plates included in this study. Normally this

**Table 3. Inter-assay CV calculated using 76 heparin plasma samples tested in 7 labs.**

| Analyte | All plates | | | Excluding 2 plates with QC results out of range | | |
|---|---|---|---|---|---|---|
| | Average inter-assay %CV | Valid CVs (n) | Valid results (n) | Average inter-assay %CV | Valid CVs (n) | Valid results (n) |
| AGP | 15.6 | 76 | 898 | 15.9 | 76 | 796 |
| CRP | 21.2 | 75 | 823 | 13.4 | 76 | 836 |
| Ferritin | 15.8 | 76 | 854 | 15.7 | 76 | 829 |
| HRP2 | 31.1 | 41 | 229 | 23.1 | 76 | 794 |
| RBP4 | 15.1 | 76 | 857 | 18.0 | 74 | 761 |
| sTfR | 25.4 | 76 | 869 | 27.9 | 46 | 219 |
| Tg | 15.8 | 76 | 901 | 14.3 | 76 | 813 |

Inter-assay CV calculated across all plates (n = 12) for each sample (n = 76); inter-assay CVs were then averaged for each analyte. CV calculations are shown with and without two plates with quality control specimen values outside the 95% confidence intervals (calculated from all plates included in this study) for multiple analytes.

would indicate a QC fail indicating the test results were not acceptable, however as this experiment was intended to assess variability across users and laboratories, results for all plates were included in subsequent analyses summarizing the intra- and inter-assay CVs irrespective of whether they passed QC or not.

Summaries of the G and H quality control sample measures, intra-assay CVs for all samples and inter-assay CVs for two quality control samples, are shown by laboratory and operator in Table 2. Values outside the limits of quantification were excluded. The maximum possible number of valid results for each lab and each operator varied because of the different numbers of plates tested. Numbers of valid results also differed because some samples had concentrations near the limits of the quantification range, calculated from mean and standard deviations for replicate standard wells on each plate. Some specimens were within range on some plates and out of range on others; thus, the number of out-of-range values is reflected in the numbers of valid results included in Table 2. For intra-assay CV, all well-to-well CVs were averaged regardless of analyte; the averages ranged from 3.1 to 8.8%, with two results above the 5% threshold coming from one laboratory. Inter-assay CVs calculated using the two quality control samples (G and H) also showed differences by laboratory.

Table 3 shows inter-assay CV by analyte for the panel of 76 heparin plasma samples run across all seven laboratories. HRP2 had the highest inter-assay CV (31.1%) but is the only analyte intended not to be interpreted quantitatively (e.g. a qualitative assay) and the test results were at the lower range of the calibration standard where the greatest variance is observed. Fig 2 shows the full distribution of results for each sample for every analyte, with the results plotted by the rank order of the mean concentration calculated using data from the plates tested in the PATH and UW laboratories (n = 3 plates per sample). The plots show the greatest scatter around these means at the lowest and highest concentrations. In general, inter-assay CVs were higher for those samples at the extremes of the analyte calibration ranges (S1 Fig).

Mean standard deviation (SD) results for each assay batch for the panel of 76 plasma samples and a measure of agreement between the results across labs assessed using Lin's Concordance Correlation Coefficient are shown in Table 4. Rather than comparing each batch in a pairwise test against all other results sets for the sample panel, a predicate set of results was derived by averaging results from batches run by the three operators at the PATH and UW labs who had the most experience with the 7-Plex assay. Lin's rho was generally high, most often $r_c \geq 0.9$, and averaging $r_c > 0.8$ for all analytes. The ferritin assay had the highest concordance of all the analytes ($r_c$ averaging 0.958), while the concordance was lowest for CRP ($r_c$

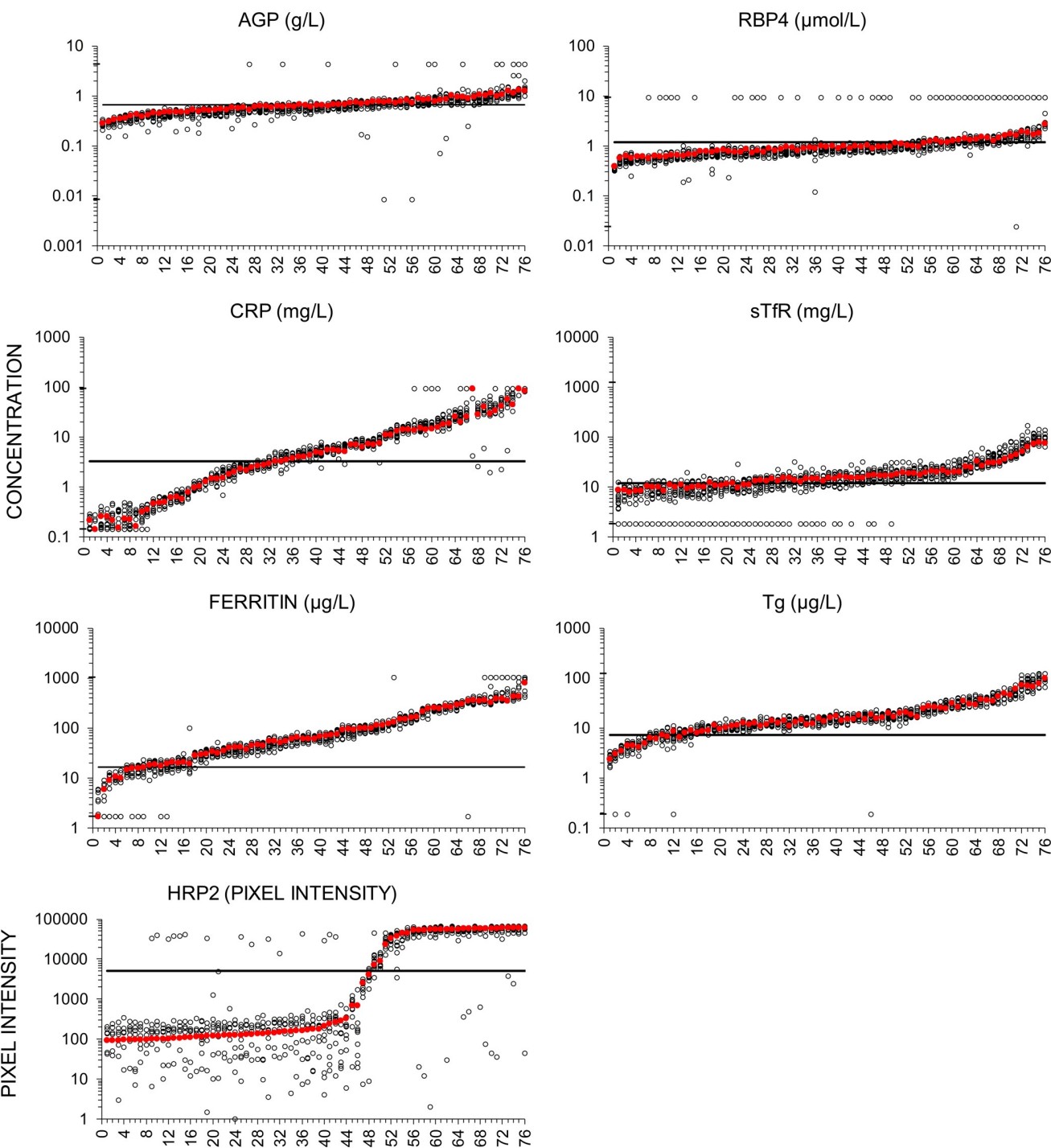

**Fig 2. Repeated measures across laboratories for a panel of 76 plasma specimens.** Open circles measured mean concentration (or pixel intensity for HRP2) of duplicate wells from a single plate. Red closed circles, mean concentration, duplicate wells run on each of 3 plates (2 PATH, 1 UW) for every sample. Results are plotted on Log10 Y axes to reveal proportional differences at lower concentrations and have been sorted by rank order of the mean concentration from 3 plates (2 PATH, 1 UW). Horizontal line, optimal 7-plex cutoff value; for HRP2, line represents approximate pixel intensity corresponding to the cutoff concentration. Cutoff values were determined by ROC curve analysis using results from the NiMaNu study as a gold-standard, and using the cutoff thresholds applied in that study [23]. Black hash marks on y-axes indicate lot-specific upper- and lower-limits of quantification (see S4 Table); values have been adjusted to account for 1:10 sample dilution used for all samples. Results out of range are plotted as the limits values noted in S4 Table. AGP, α-1-acid glycoprotein; CRP, C-reactive protein; HRP2, histidine rich protein 2; RBP4, retinol binding protein 4; sTfR, soluble transferrin receptor; Tg, thyroglobulin.

**Table 4. Mean, standard deviation, and Lin's concordance correlation coefficient for 76 heparin plasma samples tested in 7 labs.**

| LAB_ID | 1 | 1 | 1 | | 3 | 3 | 3 | 4 | 5 | 5 | 6 | 6 | 7 |
|---|---|---|---|---|---|---|---|---|---|---|---|---|---|
| operator ID | 1 | 2 | 3 | mean, batches 1, 2, 3 | 4 | 5 | 7 | 6 | 8 | 7 | 9 | 10 | 11 |
| batch ID | 1 | 2 | 3 | | 4 | 5 | 7 | 6 | 8 | 8 | 9 | 10 | 11 |
| **AGP** | | | | | | | | | | | | | |
| n | 76 | 76 | 76 | 76 | 76 | 76 | 76 | 76 | 76 | 76 | 76 | 76 | 62 |
| mean (SD), g/L | 0.690 (0.223) | 0.732 (0.267) | 0.646 (0.223) | 0.689 (0.234) | 0.633 (0.237) | 0.667 (0.254) | 0.708 (0.249) | 0.683 (0.232) | 0.594 (0.215) | 0.745 (0.434) | 0.563 (0.212) | 0.607 (0.202) | 0.425 (0.192) |
| $r_c$ | number of pairwise comparisons to mean 1, 2, 3 | | | | 0.914 | 0.931 | 0.952 | 0.966 | 0.794 | 0.724 | 0.777 | 0.876 | 0.444 |
| **CRP** | | | | | | | | | | | | | |
| n | 69 | 69 | 66 | 73 | 72 | 71 | 71 | 69 | 68 | 71 | 69 | 65 | 60 |
| mean (SD), mg/L | 8.56 (12.55) | 10.15 (14.54) | 8.00 (9.72) | 9.99 (14.64) | 9.81 (14.64) | 9.67 (12.65) | 11.72 (16.29) | 10.69 (16.50) | 10.53 (14.60) | 11.27 (15.27) | 12.31 (15.71) | 10.80 (14.19) | 5.76 (8.01) |
| $r_c$ | number of pairwise comparisons to mean 1, 2, 3 | | | | 0.897 | 0.943 | 0.977 | 0.887 | 0.959 | 0.914 | 0.962 | 0.986 | 0.463 |
| **Ferritin** | | | | | | | | | | | | | |
| n | 72 | 65 | 72 | 75 | 76 | 76 | 67 | 71 | 70 | 74 | 73 | 69 | 66 |
| mean (SD), µg/L | 142.4 (143.2) | 116.4 (114.6) | 102.4 (116.7) | 131.4 (146.5) | 123.3 (122) | 127.6 (131.3) | 144.0 (158.4) | 129.3 (124.7) | 107.0 (106.1) | 135.3 (168.1) | 151.2 (201.3) | 92.7 (98) | 95.8 (105.2) |
| $r_c$ | number of pairwise comparisons to mean 1, 2, 3 | | | | 0.935 | 0.973 | 0.942 | 0.949 | 0.995 | 0.972 | 0.902 | 0.981 | 0.976 |
| **HRP2** | | | | | | | | | | | | | |
| n | 9 | 9 | 11 | 11 | 21 | 10 | 11 | 8 | 21 | 54 | 35 | 29 | 11 |
| mean (SD), µg/L | 2.93 (2.46) | 2.50 (2.20) | 2.30 (2.47) | 2.25 (2.35) | 3.14 (2.52) | 1.8 (1.91) | 1.87 (2.24) | 1.89 (1.67) | 3.30 (2.88) | 0.34 (1.03) | 0.65 (1.48) | 0.64 (1.46) | 1.90 (2.02) |
| $r_c$ | number of pairwise comparisons to mean 1, 2, 3 | | | | 0.949 | 0.965 | 0.832 | 0.928 | 0.952 | 0.934 | 0.888 | 0.921 | 0.913 |
| **RBP4** | | | | | | | | | | | | | |
| n | 76 | 67 | 76 | 76 | 76 | 76 | 76 | 64 | 76 | 76 | 74 | 76 | 44 |
| mean (SD), µmol/L | 1.08 (0.37) | 0.89 (0.31) | 1.01 (0.43) | 1.02 (0.39) | 0.81 (0.29) | 0.85 (0.37) | 0.95 (0.39) | 0.92 (0.38) | 0.90 (0.36) | 0.96 (0.57) | 0.90 (0.4) | 0.98 (0.36) | 0.79 (0.36) |
| $r_c$ | number of pairwise comparisons to mean 1, 2, 3 | | | | 0.786 | 0.853 | 0.88 | 0.854 | 0.891 | 0.815 | 0.872 | 0.937 | 0.746 |
| **sTfR** | | | | | | | | | | | | | |
| n | 76 | 76 | 52 | 76 | 76 | 76 | 76 | 58 | 76 | 76 | 76 | 76 | 75 |
| mean (SD), mg/L | 22.2 (15.1) | 19.3 (14.7) | 20.6 (18.4) | 19.9 (15.1) | 22.2 (19.4) | 24.4 (21.5) | 17.1 (14.6) | 26.6 (24.6) | 22.4 (22.3) | 15.8 (16) | 19.7 (21.3) | 27.9 (26.7) | 24.5 (27.2) |
| $r_c$ | number of pairwise comparisons to mean 1, 2, 3 | | | | 0.942 | 0.892 | 0.971 | 0.858 | 0.899 | 0.946 | 0.932 | 0.937 | 0.728 |
| **Tg** | | | | | | | | | | | | | |
| n | 75 | 75 | 76 | 76 | 76 | 75 | 76 | 76 | 76 | 75 | 74 | 75 | 72 |
| mean (SD), µg/L | 20.7 (16.4) | 21 (17.4) | 19.6 (20) | 21.1 (19) | 20.4 (16.9) | 22.4 (21.3) | 22.1 (22.3) | 22.4 (22.3) | 18.4 (15.4) | 19.3 (18.5) | 20.3 (20.7) | 21.8 (20.1) | 21.6 (22.2) |
| $r_c$ | number of pairwise comparisons to mean 1, 2, 3 | | | | 0.979 | 0.932 | 0.948 | 0.951 | 0.922 | 0.946 | 0.87 | 0.928 | 0.955 |

Mean and standard deviation for all plasma samples (n = 76) by lab and operator. Results outside the assay limits of detection are excluded. Lin's Concordance Correlation Coefficient ($r_c$) measures agreement between a predicate result set (the mean of results for each sample across three assay batches from the most experienced users) and results from each of the remaining nine assay batches.

averaging 0.820). Concordance was notably lower across several analytes for the two batches from lab 7. Removing those batches increased the average Lin's rho for all analytes except ferritin. This result is consistent with the higher intra- and inter-assay CVs from that lab (Table 2), possibly reflecting the impact of imprecision on the concordance estimates.

## Discussion and conclusions

This study evaluates the performance of a multiplex micronutrient surveillance tool that quantifies biomarkers of vitamin A, iron, and iodine deficiency, inflammation or infection, and malaria through validation experiments to estimate precision and linearity within a single lab, along with assessing inter-laboratory reproducibility. Our within-lab validation experiments repeat and expand upon previously reported assay performance evaluations [23]. The validations described here were conducted independently in PATH's laboratory and represent an expert user's experience of assay performance characteristics across 20 repeated experiments. The intra-assay CVs were good in this validation with only one biomarker, ferritin, being slightly out of range. For most analytes and most samples, the inter-assay CV was under 15% (Table 1 and S3 Table) which is an accepted maximum inter-assay CV for ELISAs and comparable to CV's observed previously for the 7-plex assay. Because the specimens we used for replication (e.g., commercially available control specimens with and without spiking with sTfR and Tg) included values at the low and high ends of the assay range, where estimated values can be less precise, generating higher CVs were to be expected. It was not possible to produce more concentrated versions with these biomarkers without significantly diluting the other analytes. While the plasma specimens were selected from our in-house panel for the validation study based upon the highest sTfR measurements, concentrations of sTfR and Tg in each sample were still lower than their respective concentrations in the SLK. This indicated that while the range used in the validation studies was lower than the range of quantification of the 7-plex assay, they still reflected the range found in most clinical samples. Tests of assay linearity showed no evidence of systemic non-parallelism across dilutions for any analyte.

Inter-assay CVs for biomarker measurements in the panel of 76 plasma specimens measured by 11 different operators in seven laboratories averaged 20.0%, with imprecision estimates highest in the semi-quantitative HRP2 assay (31.1%), and higher than generally accepted range of error for sTfR (25.4%) and CRP (21.2%). Lin's CCC was generally high ($r_c \geq$ 0.8 for 54 of 63 comparisons), but showed the same pattern observed in CVs, with two low ($\leq$0.5 for both AGP and CRP) concordance results from one lab.

Some of the imprecision is attributable to including specimens at the very low or very high ends of the working assay ranges. For CRP in particular, most of the imprecision is due to variability at very low concentrations, all of which were below levels that indicate infection or inflammation (S1 Fig). Some individual laboratories showed variability of their results, with one laboratory generally having higher inter-assay CVs (averaging 28.1%) for quality control samples as compared to the 6 other labs (ranging from 5.6 to 14.1%). One significant difference between this laboratory and most others was the absence of an automated microtiter plate washer. It is possible that manual washing compromises assay precision; further testing is needed to confirm that speculation. When excluding these two plates and one other plate with quality control results outside a 95% confidence interval, the average inter-assay CV decreases from 20.0% to 17.2%, and decreases to 16.2% when the semi-quantitative HRP2 assay is excluded.

Because the 7-plex assay method was designed for use as a surveillance tool in LMIC and academic research facilities, the needs and challenges for assay performance are different from those associated with clinical laboratories. Assay methods used for these purposes have in the

past been selected in an ad hoc fashion as practical and financial constraints differ from those encountered in clinical laboratories. Laboratories in these settings are not routinely testing a steady number of samples as clinical laboratories do. Instead, research laboratories are engaged sporadically to assay large numbers of specimens over a short time frame. Maintaining consistency under that sporadic workflow is a particular challenge, even within laboratories. For micronutrient status surveillance, monitoring across time and space are necessary for assessment of progress, but this presents laboratory challenges. A single method that measures key indicators of nutritional status as a single tool, rather than a collection of assays from various sources assembled for individual surveys, offers opportunity to greatly improve comparability across sets of data. These benefits are realized only if the assay results are reproducible across laboratories. The results here suggest that the 7-plex can provide generally reproducible results across laboratories, with imprecision only slightly above the range acceptable for results generated within a single laboratory. They also highlight the need for including internal quality control specimens on every plate, and in cases where precise estimates are needed for values at the physiological extremes, for repeating testing at adjusted dilutions for specimens with concentrations at the margins of the assay range.

Work is ongoing to improve the performance of the 7-Plex assay and includes improvements to ferritin, RBP4, and sTfR assays. A challenge inherent to multiplex assay methods is simultaneously optimizing the assay performance for both high- and low-abundant proteins; forthcoming improvements to assay sensitivity for ferritin will allow for a change to the recommended sample dilution from 1:10 to 1:40, a change that is expected to improve precision for RBP4 assay values by bringing them closer to the middle of the reportable assay range, a previous criticism of the 7-plex [27, 28]. Similarly, changes to the sTfR assay to improve precision are underway. A new version of the 7-Plex assay with these improvements is expected within the coming year.

The project team also recognizes a need for evaluations comparable to those described here to be conducted in a field study in Low and middle income countries (LMICs). In this setting, working protocols can be developed and validated with country partners to properly implement the 7-plex into supporting nutrition research and interventions.

Our multiplex tool has to potential to reduce labor, supplies, and sample volumes traditionally required for MN screening. By demonstrating comparable performance across multiple laboratories and users we believe this assay can be a key tool in the identification of populations with key micronutrient deficiencies as well as a monitoring tool following any subsequent interventions, generating high quality reproducible data. The Quansys system is a low cost technology that could be easily implemented in laboratories in LMICs where ELISA assays are routinely used and can be applied to multiple other biomarkers and sample types beyond the 7-plex described in this work [33–36].

## Supporting information

**S1 Table. Assay precision for two specimens across 19 replicate aliquots.** Within and between tube variation in measures from 19 aliquots for each of two samples selected at random from aliquots prepared for distribution to partner labs. CVs were calculated using the Rodbard variance components model. CV for HRP2 was not calculated because results were above the upper limit of detection. CV, coefficient of variation; AGP, α-1-acid glycoprotein; CRP, C-reactive protein; HRP2, histidine rich protein 2; N/A, not available; RBP4, retinol binding protein 4; sTfR, soluble transferrin receptor; Tg, thyroglobulin.
(DOCX)

**S2 Table. The established values for the G and H controls developed for the 7-plex array.**
The expected value for each biomarker is shown in addition to the acceptable values for upper
and lower limits. AGP, α-1-acid glycoprotein; CRP, C-reactive protein; HRP2, histidine rich
protein 2; N/A, not available; RBP4, retinol binding protein 4; sTfR, soluble transferrin recep-
tor; Tg, thyroglobulin.
(DOCX)

**S3 Table. Assay precision and linearity for 7 plex array.** AGP, α-1-acid glycoprotein; CRP,
C-reactive protein; HRP2, histidine rich protein 2; N/A, not available; RBP4, retinol binding
protein 4; sTfR, soluble transferrin receptor; Tg, thyroglobulin; LK, Liquichek LK; SLK, spiked
Liquichek. After each analyte in parentheses are the LLOQ and ULOQ, respectively.
(DOCX)

**S4 Table. The upper- and lower limits of quantification for each biomarker using the
7-plex assay.** AGP, α-1-acid glycoprotein; CRP, C-reactive protein; HRP2, histidine rich pro-
tein 2; RBP4, retinol binding protein 4; sTfR, soluble transferrin receptor; Tg, thyroglobulin.
(DOCX)

**S1 Fig. Average inter-assay CV plotted against average concentration (n = 76 heparinized
plasma samples tested 12 times per sample).** Y-axes set to 100% for all analytes. One CRP
result (31 mg/L, CV = 123%) and one HRP2 result (0.2 mg/mL, CV = 177%) are not shown.
AGP, α-1-acid glycoprotein; CRP, C-reactive protein; HRP2, histidine rich protein 2; RBP4,
retinol binding protein 4; sTfR, soluble transferrin receptor; Tg, thyroglobulin.
(TIF)

## Acknowledgments

The authors would like to thank Abby Minor and Olivia Halas (both PATH) for critical review
of this manuscript. We greatly appreciate Césaire Ouédraogo (University of California Davis)
and Helen Keller International, who collaborated in the original NiMaNu study, and those
who generously provided specimens that form the basis of this evaluation.

## Disclosure statement

The findings and conclusions in this report are those of the author(s) and do not necessarily
represent the official position of the Centers for Disease Control and Prevention.

## Author Contributions

**Conceptualization:** Eleanor Brindle, David S. Boyle.

**Data curation:** Eleanor Brindle, Lorraine Lillis, Rebecca Barney, Pooja Bansil, Francisco
Arredondo.

**Formal analysis:** Eleanor Brindle, Pooja Bansil, Mindy Zhang, Christine M. Pfeiffer.

**Funding acquisition:** David S. Boyle.

**Investigation:** Lorraine Lillis, Rebecca Barney, Césaire T. Ouédraogo, Francisco Arredondo,
Mikaela K. Barker, Christina Fischer, James L. Graham, Peter J. Havel, Crystal D. Karako-
chuk, Mindy Zhang, Ei-Xia Mussai, Carine Mapango, Jody M. Randolph, Katherine
Wander.

**Methodology:** Eleanor Brindle, Sonja Y. Hess, K. Ryan Wessells, Christine M. Pfeiffer, David
S. Boyle.

**Project administration:** Eileen Murphy.

**Resources:** Rebecca Barney, Césaire T. Ouédraogo.

**Supervision:** Sonja Y. Hess, Neal E. Craft, Peter J. Havel, Crystal D. Karakochuk, Katherine Wander, Christine M. Pfeiffer, Eileen Murphy, David S. Boyle.

**Writing – original draft:** Eleanor Brindle, Lorraine Lillis, Sonja Y. Hess, K. Ryan Wessells, Neal E. Craft, Crystal D. Karakochuk, Christine M. Pfeiffer, David S. Boyle.

**Writing – review & editing:** Eleanor Brindle, Lorraine Lillis, Rebecca Barney, Sonja Y. Hess, K. Ryan Wessells, Neal E. Craft, Crystal D. Karakochuk, Mindy Zhang, Katherine Wander, Christine M. Pfeiffer, David S. Boyle.

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
