## [Decision Letter · Decision Letter 0]

5 May 2021

PONE-D-21-06971

A multicenter analytical performance evaluation of a multiplexed immunoarray for the simultaneous measurement of biomarkers of micronutrient deficiency, inflammation and malarial antigenemia

PLOS ONE

Dear Dr. Boyle,

Thank you for submitting your manuscript to PLOS ONE. After careful consideration, we feel that it has merit but does not fully meet PLOS ONE’s publication criteria as it currently stands. Therefore, we invite you to submit a revised version of the manuscript that addresses the points raised during the review process.

We look forward to receiving your revised manuscript.

Kind regards,

Guido Sebastiani

Academic Editor

PLOS ONE

Journal Requirements:

We note that one or more of the authors are employed by a commercial company: Craft Nutrition Consulting; PATH.

2.1. Please provide an amended Funding Statement declaring this commercial affiliation, as well as a statement regarding the Role of Funders in your study. If the funding organization did not play a role in the study design, data collection and analysis, decision to publish, or preparation of the manuscript and only provided financial support in the form of authors' salaries and/or research materials, please review your statements relating to the author contributions, and ensure you have specifically and accurately indicated the role(s) that these authors had in your study. You can update author roles in the Author Contributions section of the online submission form.

2.2. Please also provide an updated Competing Interests Statement declaring this commercial affiliation along with any other relevant declarations relating to employment, consultancy, patents, products in development, or marketed products, etc.  

3. We note that two NCT registries are linked in the ethics statement in the online submission form, but only one in the manuscript Methods. Please review and ensure that this is stated correctly in both the online submission form and manuscript Methods, and that ethical information for all relevant studies is included.

4. During our internal checks, the in-house editorial staff noted that you conducted research or obtained samples in another country. Please check the relevant national regulations and laws applying to foreign researchers and state whether you obtained the required permits and approvals. Please address this in your ethics statement in both the manuscript and submission information. In addition, please ensure that you have suitably acknowledged the contributions of any local collaborators involved in this work in your authorship list and/or Acknowledgements. Authorship criteria is based on the International Committee of Medical Journal Editors (ICMJE) Uniform Requirements for Manuscripts Submitted to Biomedical Journals - for further information please see here: https://journals.plos.org/plosone/s/authorship.

Reviewers' comments:

Reviewer's Responses to Questions

**Comments to the Author**

1. Is the manuscript technically sound, and do the data support the conclusions?

Reviewer #1: Yes

2. Has the statistical analysis been performed appropriately and rigorously? 

Reviewer #1: Yes

3. Have the authors made all data underlying the findings in their manuscript fully available?

Reviewer #1: Yes

4. Is the manuscript presented in an intelligible fashion and written in standard English?

Reviewer #1: Yes

5. Review Comments to the Author

Reviewer #1: The work by Brindle et al. is quite well written and extensively describes a 7-plex ELISA-based assay for the simultaneous detection of several circulating biomarkers for malaria, inflammation and malnutrition. However, some aspect of the paper could be clarified or improved:

1. In the abstract, authors say that this immunoassay can be used on both plasma and serum, however, successively authors only describe plasma analysis they performed: did you test it also in serum? If yes, please show results in a supplementary material file in order to demonstrate that both biofluid can be indifferently used or explain why they chose plasma (is it more reproducible? Is there some molecule not detectable or highly confounding in serum measurement?). If serum has not been evaluated, please remove it.

2. Since followed workflow is quite long, I strongly suggest to authors to describe it in a graphical workflow, in order to allow readers to catch key points of primary analysis and validation.

3. Patients nor controls have been described from a clinical point of view: are they comparable? Please, provide a table containing at least basic clinical characteristics of analyzed subjects (variability in molecules detection could also derive from heterogeneity of population?).

4. In Materials and Methods section authors start with "validation". However, it is impossible to start with a validation. I suggest to modify this term and substitute it with "Test" or "Setup".

5. Authors did not explain why a 7-plex only considering plasma evaluation. Could analysis of other molecules (in other biofluids) such as uric Nitrogen (evaluation of protein intake) or erythrocyte EPA (eicopentanoid acid, for omega-3 intake) render this assay more reliable and reproducible?

6. Which is the clinical relevance of this study? It is very important to demonstrate the reproducibility of this test, however, in order to render more relevant the study from a clinical point of view, I strongly suggest to authors to consider adding, in discussion section, some sentence about the clinical applicability and importance of this test. For example: this test will be potentially used prevalently in countries with a high rate of malnutrition and a high prevalence of malaria, which are not very rich countries: which is the cost of this assay? Is it applicable to villages hard to access? Which is the time for referral? Please, describe advantages and disadvantages.

7. Please, provide spelling of some abbreviation and revise some English expression.

6. PLOS authors have the option to publish the peer review history of their article (what does this mean?). If published, this will include your full peer review and any attached files.

Reviewer #1: No

---

## [Author Response · Author response to Decision Letter 0]

30 Sep 2021

June 11, 2021

We would like to note our sincere thanks to the reviewers for their providing reasoned comments for us to improve the quality of our work. We greatly appreciate this opportunity to reply to the comments raised and it is our intent to clearly address all of them in sufficient detail. Following are our responses to each comment. We have included the line numbering for where we made revisions to the original draft. 

This document is called 

Manuscript Number PONE-D-21-06971

"A multicenter analytical performance evaluation of a multiplexed immunoarray for the simultaneous measurement of biomarkers of micronutrient deficiency, inflammation and malarial antigenemia"

The work by Brindle et al. is quite well written and extensively describes a 7-plex ELISA-based assay for the simultaneous detection of several circulating biomarkers for malaria, inflammation and malnutrition. However, some aspect of the paper could be clarified or improved:

1. In the abstract, authors say that this immunoassay can be used on both plasma and serum, however, successively authors only describe plasma analysis they performed: did you test it also in serum? If yes, please show results in a supplementary material file in order to demonstrate that both biofluid can be indifferently used or explain why they chose plasma (is it more reproducible? Is there some molecule not detectable or highly confounding in serum measurement?). If serum has not been evaluated, please remove it.

Response to 1: Yes, both serum and plasma were used in the study. With serum, we used a serum-based matrix, Liquichek Immunology Control (Bio Rad) and some plasma-based clinical samples during the validation experiments while plasma (collected from the NiMaNu study) was used during the inter-lab study. This was part of a panel used in a previous paper demonstrating the assay performance (Brindle, E. et al; 2017). In our original study (Brindle, E. et al; 2014) we demonstrated comparable results with paired serum and heparinized plasma samples, confirming that either can be used. We have added some text at lines 176-178 to reflect that either sample type can be used.

2. Since followed workflow is quite long, I strongly suggest to authors to describe it in a graphical workflow, in order to allow readers to catch key points of primary analysis and validation.

Response to 2: We have added Figure 1 (See line 154-160) a flow chart depicting the three primary workstreams followed to prepare and complete the interlaboratory assessment of the 7-plex assay

3. Patients nor controls have been described from a clinical point of view: are they comparable? Please, provide a table containing at least basic clinical characteristics of analyzed subjects (variability in molecules detection could also derive from heterogeneity of population?).

Response to 3: This is a great point and highlights the importance of carrying out any validation with clinical samples, as well as repeat testing across multiple laboratories, showing that while controls can be more homogeneous, showing high reproducibility, that heterogeneity of a clinical sample could affect results. This was something we wanted to learn more about in our evaluations, both by running a sample multiple times on one plate as well as having the sample tested across different labs by different users and seeing how the assay performed. However, we do not have access to any of the clinical characteristics of the participants in the study and so cannot include them, other than to state there is inherent variability when measuring a population.

4. In Materials and Methods section authors start with "validation". However, it is impossible to start with a validation. I suggest to modify this term and substitute it with "Test" or "Setup". 

Response to 4: The reviewer is fully correct. We have moved the description of the assay set up to the beginning of the Materials and Methods section (lines 97-122), with subsequent later sections reflecting validation and inter lab assessment.

5. Authors did not explain why a 7-plex only considering plasma evaluation. Could analysis of other molecules (in other biofluids) such as uric Nitrogen (evaluation of protein intake) or erythrocyte EPA (eicopentanoid acid, for omega-3 intake) render this assay more reliable and reproducible? 

Response to 5: Our assay is intended as a tool for surveying the most common types of micronutrient deficiency, with the literature showing that plasma/serum are the best sample type for indicating this, while we have also demonstrated performance with eluates from dried blood spots. We have no doubt other biofluids could be used but have not investigated any of these. The platform used utilizes a multiplexed ELISA approach as an alternative to running multiple monoplex assays, with detection via antigens and/or antibodies and so in principle there is potential for adding and assay for eicopentanoid acid. We have demonstrated the expansion of the 7-plex to an 11-plex to support the detection of environmental enteric dysfunction and so it is feasible (Arndt M et al., 2020, PMID: 32997666).

In principle the platform can host up to 18 assays per well and thus it may be possible to develop a different assay that targets other molecules and other sample types as suggested by the reviewer. However, the inclusion of additional biomarkers for detection is dependent upon availability of appropriate assay reagents and these need to be highly compatible with the other assays already present on the array. We tried to add ELISAs to quantify vitamin D, vitaminB12 and folate but we were unable to get them to work sufficiently well as a complete array. So, while it is feasible to add other biomarkers, it can also be very challenging. A big challenge to this is also finding the funding resources to add further tests. 

A commercial component to the mass multiplexing is there needs to be a strong market demand for each of these tests to be added and this is why we chose the more common micronutrients that have biomarkers; as more tests are added, the cost goes up and Quansys are already finding that some researchers do not want malaria or iodine assays on their array. For Quansys to earn revenue (and keep making the product) there’s a need to limit the test menu. If a customer specifically wants more tests added then they could go directly to Quansys who specialize in customized arrays. We have added some text to the discussion at lines 462-464 to highlight that this technology can be applied to multiple other sample types.

6. Which is the clinical relevance of this study? It is very important to demonstrate the reproducibility of this test, however, in order to render more relevant the study from a clinical point of view, I strongly suggest to authors to consider adding, in discussion section, some sentence about the clinical applicability and importance of this test. For example: this test will be potentially used prevalently in countries with a high rate of malnutrition and a high prevalence of malaria, which are not very rich countries: which is the cost of this assay? Is it applicable to villages hard to access? Which is the time for referral? Please, describe advantages and disadvantages.

Response to 6: We have added some text in the discussion at lines 452-460 to address this. It can be noted that this is not a clinical diagnostic tool, but rather a population surveillance tool thus the purpose to look for general trends within a population as opposed to providing an individual assessment of each patient; for example assessing the effectiveness of an iron supplementation program in a region where iron deficiency is a concern.

7. Please, provide spelling of some abbreviation and revise some English expression.

Response to 7: We have revised the text to reflect the full name at the first time these abbreviations are used in order to help the reader.

Once again we would like to thank you and the reviewers for your feedback. We hope the changes we have made in response to these comments have strengthened the overall quality of manuscript and helped to clarify the primary points that we want to portray. We look forward to hearing your response.

---

## [Editor Report · Decision Letter 1]

21 Oct 2021

A multicenter analytical performance evaluation of a multiplexed immunoarray for the simultaneous measurement of biomarkers of micronutrient deficiency, inflammation and malarial antigenemia

PONE-D-21-06971R1

Dear Dr. Boyle,

We’re pleased to inform you that your manuscript has been judged scientifically suitable for publication and will be formally accepted for publication once it meets all outstanding technical requirements.

Kind regards,

Guido Sebastiani

Academic Editor

PLOS ONE
---

## [Editor Report · Acceptance letter]

25 Oct 2021

PONE-D-21-06971R1 

A multicenter analytical performance evaluation of a multiplexed immunoarray for the simultaneous measurement of biomarkers of micronutrient deficiency, inflammation and malarial antigenemia 

Dear Dr. Boyle:

I'm pleased to inform you that your manuscript has been deemed suitable for publication in PLOS ONE. Congratulations! Your manuscript is now with our production department. 

Kind regards, 

on behalf of

Dr. Guido Sebastiani 

Academic Editor

PLOS ONE